# Tobacco Use and Exposure to Environmental Tobacco Smoke amongst Pregnant Women in the United Arab Emirates: The Mutaba’ah Study

**DOI:** 10.3390/ijerph19127498

**Published:** 2022-06-18

**Authors:** Mohammed Nagdi Taha, Zaki Al-Ghumgham, Nasloon Ali, Rami H. Al-Rifai, Iffat Elbarazi, Fatima Al-Maskari, Omar El-Shahawy, Luai A. Ahmed, Tom Loney

**Affiliations:** 1College of Medicine, Mohammed Bin Rashid University of Medicine and Health Sciences, Dubai P.O. Box 505055, United Arab Emirates; mohammed.zaki@students.mbru.ac.ae (M.N.T.); zaki.alghumgham@students.mbru.ac.ae (Z.A.-G.); 2Institute of Public Health, College of Medicine and Health Sciences, United Arab Emirates University, Al Ain P.O. Box 17666, United Arab Emirates; nasloona@uaeu.ac.ae (N.A.); rrifai@uaeu.ac.ae (R.H.A.-R.); ielbarazi@uaeu.ac.ae (I.E.); fatma.am@uaeu.ac.ae (F.A.-M.); luai.ahmed@uaeu.ac.ae (L.A.A.); 3Zayed Center for Health Sciences, United Arab Emirates University, Al Ain P.O. Box 17666, United Arab Emirates; 4Tobacco, Alcohol and Drug Use Section, Department of Population Health, New York University Grossman School of Medicine, New York, NY 10016, USA; omar.elshahawy@nyulangone.org; 5Division of Global Health, New York University School of Global Public Health, New York, NY 10003, USA

**Keywords:** birth, cohort, early-life exposures, indoor air pollution, mother, pregnancy, tobacco smoke pollution, tobacco use, United Arab Emirates

## Abstract

Self-reported tobacco use is high in the male adult Emirati population (males ~36% vs. females ~3%); however, there are minimal data on tobacco use or exposure to environmental tobacco smoke (ETS) during pregnancy in the United Arab Emirates (UAE). This study investigated the prevalence of, and factors associated with, tobacco use and exposure to environmental tobacco smoke (ETS) amongst pregnant women in the UAE. Baseline cross-sectional data were analysed from the Mutaba’ah Study. Expectant mothers completed a self-administered questionnaire collecting sociodemographic information, maternal tobacco use, and ETS exposure during antenatal visits at three hospitals in Al Ain (UAE; May 2017–February 2021). Amongst 8586 women included in the study, self-reported tobacco use during pregnancy was low (0.7%), paternal tobacco use was high (37.9%), and a third (34.8%) of expectant mothers were exposed to ETS (28.0% at home only). Pregnant women who were employed (adjusted odds ratio (aOR): 1.35, 95% confidence interval (CI): 1.19–1.52), with childbirth anxiety (aOR 1.21, 95% CI 1.08–1.36), and with an increased number of adults living in the same household (aOR 1.02 95% CI 1.01–1.03) were independently more likely to be exposed to ETS. Pregnant women with higher education levels (aOR 0.84, 95% CI 0.75–0.94) and higher gravidity (aOR 0.95, 95% CI 0.92–0.99) were less likely to be exposed to ETS. Public health efforts targeting smoking cessation amongst husbands and promoting smoke-free homes are warranted to help reduce prenatal ETS exposure in the UAE.

## 1. Introduction

Globally, tobacco smoking has consistently been associated with many illnesses and unfavourable health outcomes such as cardiovascular- and cancer-related morbidity and mortality [1]. Nonetheless, the prevalence of tobacco smoking remains high, with an estimated 19% of the world’s adult population using tobacco products: 33% amongst males and 6% amongst females [2]. Similarly, a third of male and female non-smokers and 40% of children were estimated to be exposed to environmental tobacco smoke (ETS) worldwide in 2004 [3]. The highest proportion of ETS exposure was in Europe and the Western Pacific, with more than 50% of the population exposed [3]. In contrast, estimates from Region B of the Eastern Mediterranean (including Bahrain, Iran, Jordan, Kuwait, Lebanon, Libya, Oman, Qatar, Saudi Arabia, Syria, Tunisia, and the United Arab Emirates; UAE) showed that more than a third of children and one quarter of males and females were exposed to ETS [3]. However, these global and regional estimates are most likely outdated, as evidence from some countries has indicated that the prevalence of ETS exposure has decreased. For instance, the prevalence of ETS in the United States decreased from 88% to 25% over a 26-year period [4]. Following the implementation of smoke-free policies, several European countries experienced decreases in ETS exposure between 2007–2012, with reductions up to 80–90% in workplaces and public places [5,6,7]. Moreover, a recent review indicated that smoke-free legislation had a positive effect on promoting private smoke-free settings [8]. Despite the implementation of smoke-free legislation in many European countries, recent data from the Global Adult Tobacco Survey (2009–2013) reported that approximately 50% of the one billion children (<15 years) in 21 countries were exposed to ETS at home, ranging from ~5% in Panama to 16% in Qatar and 79% in Indonesia [9]. Bangladesh, China, India, Indonesia, and the Philippines accounted for approximately 85% of ETS-exposed children, and the prevalence of ETS exposure was higher in countries with higher adult smoking rates [9]. Overall, a significant proportion of the global population is still exposed to ETS, especially at home; however, there is a dearth of ETS exposure data for the Middle East region.

Exposure to ETS is an important public health issue, as there is growing evidence that ETS exposure leads to adverse health outcomes such as an increased risk of overall cancers in non-smokers, particularly the risk of lung cancer and breast cancer in women [10], and increased risk of coronary heart disease (CHD) by 25–30% [11]. In 2017, the Global Burden of Disease Study estimated that the disease burden due to ETS exposure at home in European countries was 24,000 deaths (0.46% of total deaths) and 526,000 disability-adjusted life years (DALYs; 0.36% of total DALYs) in all adults, mainly due to chronic obstructive pulmonary disease and ischemic heart disease [12]. Individuals with early childhood exposure to ETS have a higher risk of developing respiratory illnesses such as asthma, chronic bronchitis, and shortness of breath later in life [13,14]. Prenatal ETS exposure has been shown to increase the risk of premature birth, shorter baby length, smaller head circumference, and congenital defects [15,16]. Similarly, postnatal ETS exposure in children has been associated with adverse effects on weight and height throughout childhood [15,16].

Previous work in Brazil, Finland, Portugal, Serbia, and the United States has shown that maternal age, lower educational level, and lower socioeconomic status was associated with prenatal tobacco use [17,18,19,20]. High levels of anxiety, depression, and perceived stress were also related to prenatal tobacco use and ETS exposure [21,22,23]. In Portugal, lower educational level was significantly associated with both higher smoking prevalence and ETS exposure and lower smoking cessation during pregnancy [24]. A recent pilot study on a convenient sample (*n* = 517) of Emirati adults showed that the prevalence of self-reported tobacco use amongst the Emirati adult population was 36% in men and 3% in non-pregnant women [25]. However, urine cotinine levels (>200 ng/mL) suggested even higher levels of tobacco use amongst male (42%) and female (9%) Emirati adults [25]. Despite the reported high prevalence of tobacco use in the Emirati population, there is a lack of data on the prevalence of ETS exposure in the UAE population, especially amongst pregnant women. Moreover, it is important to conduct research to identify the sociodemographic factors associated with parental tobacco use and ETS exposure during pregnancy. Such information might be useful when designing culturally relevant smoke-free home interventions to decrease prenatal tobacco use and exposure. Therefore, this study aimed to investigate the prevalence of, and factors associated with, tobacco use and exposure to ETS amongst pregnant women in the UAE.

## 2. Materials and Methods

### 2.1. Study Design, Setting and Participants

This cross-sectional analysis uses baseline data from the Mutaba’ah Mother and Child Health Study. The Mutaba’ah (which means *‘Follow-Up’* in Arabic) Study methods have been described in detail elsewhere [26]. In brief, Mutaba’ah is a prospective cohort study that aims to systematically recruit 10,000 pregnant women during their antenatal care visits at three major hospitals in the city of Al Ain, Abu Dhabi, UAE. All pregnant women from the Emirati population, who are at least 18 years old, resident in Al Ain, and able to provide informed consent, and their new-borns were eligible to participate in the study. The mothers and their offspring will be followed up until the child turns 16 years of age using questionnaires, medical record extractions, and interviews. At recruitment, women answered a self-administered short questionnaire (SQ) on an electronic tablet during their antenatal visits. The SQ includes 67 questions asking the women about various psycho-social demographics of life, previous pregnancies, and behaviours during her current pregnancy. The current cross-sectional analysis includes data from the SQ for participants recruited from May 2017 to February 2021. The study was approved by the Research Ethics Committees of the United Arab Emirates University (ERH-2017-5512), Al Ain Hospital (AAHEC-03-17-058), and Tawam Hospital (IRR–494), and was in complete agreement with the Declaration of Helsinki. All participants provided written informed consent.

### 2.2. Variables and Measurement

The socio-demographic and pregnancy-related characteristics examined included age of the mother (years), educational attainment (illiterate never attended school, primary, secondary, vocational/diploma, bachelors, masters, doctorate), employment status (seeking employment, student, housewife, employed, self-employed, retired), number of adults living in the same house, childbirth anxiety, perceived social support, gestational age (weeks), number of previous pregnancies (gravidity), and number of children (parity). Childbirth anxiety was assessed with the question *‘Do you worry about the upcoming birth?’* on a four-point Likert scale *‘Yes, quite a lot’*, *‘Yes, sometimes’*, *‘No, not much’*, or *‘No, not at all’*. Perceived social support was assessed with the question *“Do you feel that you have enough people in your life to count on when you need anything?”* on a four-point Likert scale *‘No, definitely not enough’*, *‘No, Not enough’*, *‘Yes, enough’*, or *‘Yes, Definitely enough’*. Current and pre-pregnancy maternal smoking habits were assessed with the questions *‘Did you smoke before this current pregnancy?’* and *‘Do you smoke now during this pregnancy?’* with the options of *‘Never’*, *‘Occasionally’*, or *‘Regularly’*. Current and pre-pregnancy paternal smoking habits were assessed with the questions *‘Did your husband smoke before this current pregnancy?’* and *‘Does your husband smoke now during this pregnancy?’* with the options of *‘Never’*, *‘Occasionally’*, or *‘Regularly’*. ETS exposure was assessed with the question *‘Are you exposed to passive smoking (someone smokes next to you) at home or work?* with the options of *‘No’*, *‘Yes at work’*, *‘Yes at home’*, or *‘Yes, at both places’*.

Maternal highest education was classified into: (i) high school and below or (ii) more than high school. Responses on employment status were recorded as: (i) employed—employed/self-employed or (ii) housewife/student/unemployed/retired.

### 2.3. Statistical Analyses

Descriptive statistics were performed to explore the distribution of the study sample characteristics. Continuous variables were presented as means with standard deviations, whilst categorical variables were presented as counts (percentages). Continuous variables with a normal distribution were compared using the Student’s *t*-test whilst categorical data were compared using the Pearson Chi-square test. Univariate and multivariate logistic regression models were used to quantify the association between different socio-demographic and pregnancy-related characteristics and maternal/paternal tobacco use and ETS exposure during pregnancy. Backward stepwise multivariate analyses were performed with a removal criterion of *p*-value equal to 0.10. Crude and adjusted odds ratios (aOR) with 95% confidence intervals (CI) were calculated. Statistical analyses were performed using Stata 15.1 (Stata Corp, College Station, TX, USA). A *p*-value less than or equal to 0.05 defined statistical significance.

## 3. Results

During the study period, a total of 8586 pregnant women were recruited between May 2017 and February 2021. Only 0.7% (*n* = 61) of mothers and more than a third of fathers (37.9%, *n* = 3255) reported tobacco use during pregnancy, whilst approximately 1.5% (*n* = 131) of mothers and 41.8% (*n* = 3584) of fathers smoked before pregnancy (Figure 1), according to reports by the women. From the study population, there were 2985 (34.8%) pregnant women exposed to ETS during their pregnancy (Table 1). Pregnant women exposed to ETS were significantly younger with a lower parity, and a greater proportion were employed, educated at high school or lower levels, were primipara, reported childbirth anxiety, and lived in higher occupancy households compared to pregnant women not exposed to ETS (all *p*-values ≤ 0.05; Table 1).

With reference to the location of ETS exposure, approximately 0.8% (*n* = 68) of women reported that they were exposed to smoke at work alone, whilst 28.0% (*n* = 2403) of women reported being exposed only at home, and 6.0% (*n* = 514) of women were exposed at both their home and the workplace (Figure 1).

Table 2 shows the crude and adjusted associations between multiple socio-demographic factors and ETS exposure. In the crude model, women who were employed (OR: 1.20, 95% CI 1.09–1.32), pregnant for the first time (primipara) (OR: 1.24, 95% CI 1.11–1.39), and those who reported childbirth anxiety (OR: 1.22, 95% CI 1.11–1.35) were more likely to be exposed to ETS. Pregnant women who were older (OR: 0.98, 95% CI 0.97–0.99), achieved higher levels of education (OR: 0.89, 95% CI 0.81–0.97), and had greater gravidity (OR: 0.93, 95% CI 0.91–0.95) had lower odds of ETS exposure. After controlling for potential confounding factors in the adjusted model, pregnant women who were employed (aOR 1.35, 95% CI 1.19–1.52), reporting childbirth anxiety (aOR 1.21, 95% CI 1.08–1.36), or living with more adults (aOR 1.02, 95% CI 1.01–1.03) were significantly more likely to be exposed to ETS (Table 2). Pregnant women with higher levels of education (aOR 0.84, 95% CI 0.75–0.94) and higher gravidity (aOR 0.95, 95% CI 0.92–0.99) were less likely to be exposed to ETS (Table 2).

Table 3 shows the associations between husbands who smoked during pregnancy in comparison to those who did not smoke during pregnancy as reported by the women. In both the univariate and multivariate logistic regression model, women who were employed (OR: 1.16, 95% CI 1.06–1.28; aOR: 1.28, 95% CI 1.14–1.45), those reporting childbirth anxiety (OR: 1.19, 95% CI 1.09–1.32; aOR: 1.24, 95% CI 1.10–1.38), and those who lived with more adults (OR: 1.02, 95% CI: 1.01–1.03 aOR: 1.02, 95% CI 1.01–1.04) were associated with a husband who smoked during pregnancy. Conversely, women who were more educated were less likely to have husbands who smoked during pregnancy (OR: 0.82, 95% CI: 0.75–0.89, aOR: 0.74, 95% CI 0.66–0.83).

There were no statistically significant relationships between maternal factors and paternal tobacco cessation during the current pregnancy; however, the association between women being less worried about giving birth and their husbands stopping tobacco use during pregnancy was significant in the multivariate model (aOR: 0.75, 95% CI 0.58–0.98) (Table 4).

## 4. Discussion

This large population-based study provides the first estimates on the prevalence of prenatal tobacco use and ETS exposure among pregnant women from the Emirati population in the UAE. Less than 1% of pregnant women and more than a third of their husbands were self-reported by the women to use tobacco during the current pregnancy whilst 42% of husbands and ~2% of women reported tobacco use prior to the pregnancy. Approximately 35% of the women enrolled in the study were exposed to ETS, predominantly at home. Pregnant women who were employed, experiencing childbirth anxiety, and living in higher occupancy households were significantly more likely to be exposed to ETS. However, pregnant women with higher levels of education and higher gravidity were less likely to be exposed to ETS.

Currently, there is a lack of periodic tobacco use surveillance data for the UAE; however, the prevalence estimates in our population-based study are similar to those in other neighbouring Gulf countries. Qatar participated in the 2013 World Health Organisation Global Adult Tobacco Survey, which recruited a representative sample (*n* = 4279) of Qataris and reported that 22.0% of males and 0.6% of females smoked any tobacco product, with 18.5% of males and 0.3% of females reported smoking cigarettes, 5.3% of males and 0.4% of females used shisha (water pipe), and 1.1% of males and 0.2% of females smoked midwakh pipes [27]. Self-reported tobacco use amongst pregnant women is also low in African countries. A recent study utilized data from the Demographic and Health Surveys conducted in 31 sub-Saharan Africa countries between 2008 and 2017 (*n* = 44,715 pregnant women aged 15–49 years) and reported that the overall prevalence of tobacco use amongst pregnant women was 1.9%, ranging from 11.0% in Madagascar to ~0.3% in Cameroon, Ghana, Nigeria, Togo, and Zimbabwe, with these latter countries reporting similar prenatal tobacco use estimates as the UAE [28]. Interestingly, only 1.3% of pregnant Muslim women reported smoking compared to 2.0% of pregnant Christian women, and 4.4% of pregnant women from other religions in sub-Saharan Africa [28]. The Emirati population in the UAE are all Muslim and there may be religious and cultural factors that influence tobacco use amongst women, especially during pregnancy, in our sample.

Overall, the prevalence of prenatal tobacco use in North American and European countries seems to be higher than in Gulf countries; however, there are some populations within these countries with similar prevalence estimates to the UAE. In 2016, 7.2% of women who gave birth in the United States (US) smoked cigarettes during pregnancy, with non-Hispanic American Indians or Alaska Native women reporting the highest prevalence of smoking during pregnancy (16.7%), whilst non-Hispanic Asian women had the lowest (0.6%) [29]. Another study in the US analysed data from all pregnant women (18–54 years; *n* = 726) in waves 1 and 2 of the Population Assessment of Tobacco and Health Study (2014–2015) and reported that 6.1% of pregnant women smoked during pregnancy, 23.0% reported ETS, and 11.8% reported both maternal smoking during pregnancy and ETS exposure [30]. Similarly, a prospective cohort study in France from 2010–2011 followed pregnant women (*n* = 642) until delivery and measured cotinine concentration in the meconium of offspring to determine maternal tobacco use status [25]. A fifth (21.2%) and 17.2% of pregnant women in France self-reported tobacco use during their pregnancy (20.0% cotinine positive) and third trimester, respectively, with 43.0% exposed to ETS from their partner [31]. A retrospective chart review of 856 maternity records in Reykjavik (Iceland) in 2006–2013 showed that 12.2% of women smoked at the first visit, 63 stopped during early pregnancy, and 45 (5.3%) continued smoking throughout the pregnancy [32]. The prevalence of prenatal smoking in France and Iceland were considerably higher than the 1.5% and 0.7% of the current cohort that self-reported smoking before and during pregnancy, respectively. In summary, our study revealed that more than a third of pregnant women from the Emirati population were exposed to ETS, which is lower than China (60.5%) [33] and France (43.0%) [31], but higher than the United States (23.0%) [30].

There is a paucity of population-based estimates on the prevalence of, and factors associated with, maternal tobacco use and prenatal ETS exposure. Our study found that employment, childbirth anxiety, and a larger number of adults in the same household were positively associated, and higher maternal education and increased gravidity were negatively related to ETS exposure. Similar to the present study finding an association between lower levels of education and ETS exposure, the US Population Assessment of Tobacco and Health Study (2014–2015) also reported that women who smoked during pregnancy or non-smoking pregnant women exposed to ETS were more likely to be unmarried and have lower levels of education compared to women not exposed to tobacco smoke during pregnancy [30]. Similarly, data from the Demographic and Health Surveys conducted in 31 sub-Saharan Africa countries (2008–2017) reported that the prevalence of prenatal tobacco use decreased with increasing education (no education 2.7%, primary 1.8%, secondary 0.9%, tertiary 0.2%) and wealth index (poor 2.7%, middle 1.7%, rich 0.9%; [28]). In France, the risk of tobacco use decreased with increasing paternal age and parental education or socioeconomic status, if the mother was employed during pregnancy, satisfied with her marital status, and the pregnancy was wanted [31]. These findings suggest that sociodemographic conditions, particularly education, play an important role in specific prenatal health behaviours such as tobacco use. A recent review reported that adults in the US with the lower educational levels of a graduate record equivalency or less than high school had the highest smoking rates [34]. Another study amongst 619 pregnant women in Portugal reported that a lower educational level was significantly associated with both higher smoking prevalence and exposure to ETS and lower smoking cessation [24]. Finally, a study of 179 paediatric patients (mean (SD) age = 7.9 (4.3) years) in the United States (2016–2019) who lived with at least one smoker reported that the highest urinary cotinine levels were recorded in children whose parents had a lower education [35]. It is plausible to suggest that lower levels of education might be associated with lower levels of health literacy about the negative health effects of tobacco that lead to the initiation of tobacco use during adolescence and early adulthood that continues into the prenatal period as either paternal ETS exposure and/or prenatal tobacco use. Moreover, higher maternal educational levels and concomitantly greater health literacy related to the negative health effects of prenatal ETS exposure might lead to a greater likelihood of the pregnant woman maintaining a tobacco-free home during her pregnancy.

It is unclear why higher gravidity was related to a reduced likelihood of ETS exposure; however, it might be that multigravida and multiparous women have a greater awareness around the negative health effects of prenatal ETS exposure and, hence, restrict their husband’s or family’s tobacco use in the household. Similarly, older siblings of the unborn child might have received tobacco-related health education at school and exert an influence on the household norms of tobacco use and exposure. Unfortunately, we do not have any data to further explore these suppositions. Maternal employment, childbirth anxiety, and higher household occupancy could possibly be proxy markers of socioeconomic status. As such, this could possibly explain the independent relationship between these factors and prenatal ETS exposure, as previous research in both North and South American, and European maternal cohorts reported that these factors and socioeconomic status were associated with parental tobacco use and ETS exposure [17,18,19,20,21,22,23]. Logically, it is plausible that if women are employed then they are more likely to be exposed to ETS in or around the workplace and the likelihood of ETS exposure at home increases with higher household occupancy. Due to the cross-sectional nature of the baseline cohort data, the direction of the relationship between childbirth anxiety and ETS exposure is unclear, as it may be ETS exposure causing childbirth anxiety.

Sub-optimal wellbeing also seems to be an important factor associated with tobacco use, as data from the US National Survey on Drug Use and Health (2005–2014) found that pregnant women with major depressive episodes were 2.5 times more likely to smoke cigarettes than non-smoker pregnant women [36]. Similarly, the risk of tobacco use amongst pregnant women in France was increased if the mother reported spending free time alone rather than with her family or friends, with the number of previous children, if the father was a current tobacco smoker, and if the mother reported a history of depressive disorders [31]. Our study found a positive relationship between childbirth anxiety and ETS exposure but not prenatal tobacco use.

It is noteworthy to mention the independent lower odds of women reporting fear of childbirth when their husbands quit smoking. Women who have more agency in restricting their husband’s tobacco use have done so using narratives about the importance of child health in other countries, such as China [37]. This should be incorporated in the UAE as well. The impact on paternal smoking cessation on the mental health of the women might do more than can be elucidated in this study. Further research is needed to explore the impact of ETS and spouses’ quitting smoking on both delivery and child outcomes.

Environmental tobacco smoke exposure is an important public health issue in the Middle East. Limited available earlier work has reported that tobacco use is high amongst the male adult population in the region [25,27] and, consequently, a large proportion of female adults, at least in Jordan and Gaza, are exposed to ETS [38,39,40]. To our knowledge, this is the largest population-based study in the Middle East to investigate the prevalence of, and factors associated with, prenatal tobacco use and ETS exposure. The study minimized selection bias by recruiting a large representative sample of the population from both public and private hospitals. There is no difference in healthcare access between pregnant women attending these three hospitals and those who use other institutions, as the Emirati population has full health insurance allowing them to have the same level of healthcare at any health facility. Hence, a representative sample of the Emirati population in Al Ain was recruited from these three hospitals, permitting the generalization of the findings to the population of pregnant Emirati women in Al Ain. Recall bias was minimized using a standardized questionnaire collecting information during the current pregnancy. However, one of the major limitations in this study was that both prenatal and husband tobacco use was self-reported by the women, and this was used to determine exposure status. A previous pilot study in non-pregnant women reported discrepancies between self-reported (3%) and objectively assessed tobacco use (9% based on urinary cotinine levels >200 ng/mL) in Emirati women [25]. As such, future studies in pregnant women might consider collecting both subjective and objective measures of tobacco use in pregnant women. Similarly, objective estimates of ETS exposure amongst non-smoking pregnant women would allow for more accurate and reliable tracking of temporal trends in ETS exposure. For example, recent data from eight National Health and Nutrition Examination Surveys (two-year cycles from 2003 to 2018) in the United States using serum cotinine concentrations among non-smokers as a measure of ETS showed that ETS exposure decreased by 11% every two years in all demographic groups [41]. Such longitudinal data are useful for evaluating the impact of smoke-free laws and policies on reducing ETS exposure in workplaces, public places, and homes.

A further limitation is that prenatal and paternal tobacco use was assessed as a broad concept of smoking tobacco, regardless of the type of tobacco or device used to smoke tobacco. Recent studies from the UAE [25] and Qatar [27] have shown that various forms of tobacco smoking are popular, including cigarettes, shisha smoked in a water pipe, and dokha smoked in a midwakh pipe. Future work would do well to explore possible differences in exposure levels by tobacco type and smoking device. The presented data used baseline cross-sectional data from a large ongoing prospective cohort study and, as in other cross-sectional analyses, the reported associations do not indicate causality. Finally, the cohort recruitment and completion of the short questionnaire for the data contained in this study took place between May 2017 and February 2021, which included part of the COVID-19 pandemic. Although not possible with our study dataset, cohorts with repeated measures of tobacco use before and during the COVID-19 pandemic would do well to explore temporal changes in parental tobacco use and/or prenatal ETS exposure.

### Practical Implications

Baseline data from the largest population-based prospective cohort study (viz. Mutaba’ah Study) in the UAE shows that ETS exposure occurs predominantly at home via paternal or familial tobacco use. Study findings also suggest that there is an increased likelihood of ETS exposure with higher household occupancy and that prenatal ETS exposure is associated with childbirth anxiety. Previous research has shown that ETS exposure can lead to breast cancer in women [10] and increase the risk of premature birth, babies born small for gestational age, and congenital defects [15,16]. Therefore, it is pertinent to suggest that multi-level public health interventions in the UAE are required to (i) prevent young males and females from initiating tobacco use during adolescence, (ii) improve tobacco cessation services and quit rates amongst young adults, especially husbands, and (iii) develop and implement tobacco-free home policies for households with pregnant women and/or young children.

## 5. Conclusions

Previous work using smaller convenient samples has shown a high prevalence of self-reported tobacco use amongst the male Emirati population. Our study is the first to describe tobacco use and ETS exposure amongst a large representative sample of pregnant Emirati women and their husbands in the UAE. Approximately a third of pregnant women in the UAE are exposed to ETS, and this is predominantly at home via paternal or familial tobacco use. Less than 1% of pregnant women in our cohort self-reported tobacco use, and this could be underestimated due to social desirability bias. Future prospective maternal and child health studies would do well to include objectively assessed tobacco use to explore the agreement between self-reported and objective measures of tobacco use and exposure. Public health efforts to improve smoking cessation amongst young male adults, especially husbands, and increase the proportion of smoke-free homes are warranted to help to reduce ETS exposure during pregnancy in the UAE

## Figures and Tables

**Figure 1 ijerph-19-07498-f001:**
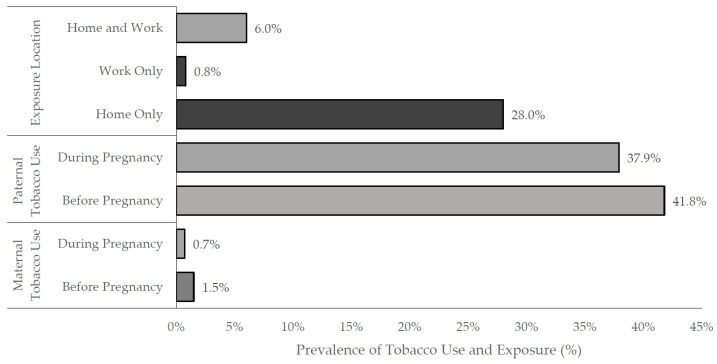
Prenatal tobacco exposure and self-reported parental tobacco use amongst 8586 pregnant women in Al Ain, UAE. The Mutaba’ah Study.

**Table 1 ijerph-19-07498-t001:** Demographic characteristics of 8586 pregnant women stratified by their ETS exposure in Al Ain, UAE: The Mutaba’ah Study.

Variable	Exposed to ETS	Not Exposed to ETS	*p*-Value
*n*	2985 (34.8%)	5601 (65.2%)	
Age, mean ± SD	30.4 ± 6.0	31.2 ± 6.1	<0.001
Number of pregnancies, mean ± SD	2.4 ± 2.0	2.7 ± 2.2	<0.001
Education			0.012
*High school and below*	1764 (60.0%)	3134 (57.2%)	
*More than High school*	1176 (40.0%)	2348 (42.8%)	
Employment			<0.001
*Employed*	1063 (36.3%)	1760 (32.1%)	
*Not employed*	1869 (63.7%)	3718 (67.9%)	
First Pregnancy			<0.001
*Yes*	670 (22.8%)	1061 (19.1%)	
*No*	2275 (77.2%)	4484 (80.9%)	
Childbirth anxiety			<0.001
*Yes*	2041 (70.1%)	1853 (34.3%)	
*No*	870 (29.9%)	3.551 (65.7%)	
Perceived social support			0.2
*Yes*	2630 (90.4%)	4931 (91.2%)	
*No*	281 (9.6%)	476 (8.8%)	
Total adults in the same house	6.1 ± 4.4	5.8 ± 4.3	<0.001
House ownership			0.31
*Yes*	2373 (82.7%)	4349 (81.8%)	
*No*	497 (17.3%)	969 (18.2%)	
Planned pregnancy			0.07
*Yes*	1588 (53.9%)	3080 (56.0%)	
*No*	1358 (46.1%)	2420 (44.0%)	

SD: standard deviation. Continuous variables were compared using the Student’s *t*-test whilst categorical data were compared using the Pearson Chi-square test.

**Table 2 ijerph-19-07498-t002:** Crude and multivariate regression model results on the associations between socio-demographic and pregnancy-related factors and self-reported exposure to ETS in pregnant women in Al Ain, UAE. The Mutaba’ah Study.

ETS—(Exposed, Reference: Not Exposed)	OR	* Adjusted OR
Age	0.98 (0.97–0.99)	0.99 (0.98–1.00)
Education	0.89 (0.81–0.97)	0.84 (0.75–0.94)
Employment	1.20 (1.09–1.32)	1.35 (1.19–1.52)
Gravidity	0.93 (0.91–0.95)	0.95 (0.92–0.99)
First pregnancy	1.24 (1.11–1.39)	1.04 (0.88–1.22)
Childbirth anxiety	1.22 (1.11–1.35)	1.21 (1.08–1.36)
Social support	0.90 (0.77–1.05)	0.91 (0.76–1.09)
Total adults in same household	1.02 (1.01–1.03)	1.02 (1.01–1.03)

* Adjusted models included all covariates listed in the table.

**Table 3 ijerph-19-07498-t003:** Crude and multivariate regression model results on the associations between socio-demographic and pregnancy-related factors and self-reported husband tobacco use amongst pregnant women in Al Ain, UAE. The Mutaba’ah Study.

Husband Who Used Tobacco during Participant’s Pregnancy (Reference: Husbands Who Did Not Smoke during Pregnancy)	OR	* Adjusted OR
Age	0.98 (0.97–0.99)	0.99 (0.98–1.00)
Education	0.82 (0.75–0.89)	0.74 (0.66–0.83)
Employment	1.16 (1.06–1.28)	1.28 (1.14–1.45)
Gravidity	0.95 (0.93–0.98)	0.96 (0.92–1.00)
First pregnancy	1.11 (1.00–1.24)	0.94 (0.80–1.10)
Childbirth anxiety	1.19 (1.09–1.32)	1.24 (1.10–1.38)
Social support	0.85 (0.73–0.99)	0.84 (0.70–1.00)
Adults in same household	1.02 (1.01–1.03)	1.02 (1.01–1.04)

* Adjusted models included all covariates listed in the table.

**Table 4 ijerph-19-07498-t004:** Crude and multivariate regression model results on the associations between socio-demographic and pregnancy-related factors and self-reported husband tobacco cessation amongst pregnant women in Al Ain, UAE. The Mutaba’ah Study.

Husbands Who Stopped Using Tobacco during the Current Pregnancy (Reference: Husbands Who Were Smoking before and during Pregnancy)	OR	* Adjusted OR
Age	0.99 (0.97–1.01)	0.99 (0.96–1.02)
Education	1.18 (0.95–1.46)	1.21 (0.93–1.58)
Employment	0.91 (0.73–1.13)	0.94 (0.71–1.26)
Gravidity	0.97 (0.92–1.02)	1.01 (0.92–1.11)
First pregnancy	1.21 (0.94–1.55)	1.36 (0.95–1.97)
Childbirth anxiety	0.80 (0.64–1.00)	0.75 (0.58–0.98)
Social support	1.33 (0.90–1.96)	1.40 (0.87–2.26)
Total people living in household	0.99 (0.98–1.00)	0.97 (0.94–1.00)

* Adjusted models included all covariates listed in the table.

## Data Availability

The data presented in this study are available on request from the Mutaba’ah Study.

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
