# Peer review of "Tobacco Use and Exposure to Environmental Tobacco Smoke amongst Pregnant Women in the United Arab Emirates: The Mutaba’ah Study"

_ijerph, 2022, doi:10.3390/ijerph19127498_

Round 1

Reviewer 1 Report

Very good work, with important applications in public health.

Just these little spelling details:

-Abstract Line 22: “were” instead of “was

-Statistical Analyzes Line 119: “were” instead of “was

-Results, Table 1 – In first pregnancy, accommodate the word “No”

-Discussion Line 239 – A parenthesis is missing after 60.5%

- Discussion Line 240 – Add a comma instead of period, after [17].

Author Response

Reviewer 1

Comments and Suggestions for Authors

Very good work, with important applications in public health.

Author Response: We thank Reviewer 1 for taking the time to review our manuscript and providing supportive feedback.

Just these little spelling details:

-Abstract Line 22: “were” instead of “was”

Author Response: Thank you for identifying our grammatical error. We have corrected this in the manuscript.

-Statistical Analyzes Line 119: “were” instead of “was”

Author Response: Thank you for highlighting our grammatical error. We have corrected this in the manuscript.

-Results, Table 1 – In first pregnancy, accommodate the word “No”

Author Response: Thank you. We have repositioned “No” in “First Pregnancy” in Table 1.

-Discussion Line 239 – A parenthesis is missing after 60.5%

Author Response: Thank you for identifying our omission. We have added the closed parenthesis in the manuscript.

- Discussion Line 240 – Add a comma instead of period, after [17].

Author Response: We have rectified this error in the manuscript. Thank you.

Reviewer 2 Report

The paper was well constructed and well written. 

I have following comments;

  1. In the introduction, please clearly state why it is important to explore factors associated with pregnancy women's exposure to environmental tobacco smoke (ETS), and factors associated with husband tobacco use amongst pregnant women, and husband tobacco cessation.  
  2. It is important to present theoretical assumptions. Please discuss why those socio-demographic of participants and pregnancy-related factors (such as age, education, employment, gravidity, first pregnancy, childbirth anxiety, and  Social support ) could affect ETS, husband tobacco use amongst pregnant women, and husband tobacco cessation.  Relevant theories and studies must be used to support the assumptions. 
  3. Regarding the methodology, please indicate how those questions used for data collection were developed from.
  4. Have you tested reliability and validity of your research tool (measurement)?
  5. Regarding the table 1, please indicate a type of statistic used for testing statistical difference of variable scores. Those statistics are such as t-test, ANOVA, Chi-square, etc.
  6. For the discussion, the interpretations of the results were not clearly. please provide logical reasons why pregnant women who were employed, had childbirth anxiety, and had an increased number of adults living in the same household were i more likely to be exposed to ETS.
  7. I also do not understand why pregnant women with higher education levels and higher gravidity were less likely to be exposed to ETS. Please provide concrete reasons.
  8. Please also provide reasons how those significant variables could be associated with  husband tobacco use amongst pregnant women, and husband tobacco cessation.  

Author Response

Reviewer 2

The paper was well constructed and well written. 

Author Response: Thank you for your thorough review, critique, and suggestions. We believe that your insightful feedback has helped us to improve our manuscript and we hope that you think the revised version makes a useful addition to the sparse literature on environmental tobacco smoke exposure amongst pregnant women in the Middle East region.

I have following comments;

In the introduction, please clearly state why it is important to explore factors associated with pregnancy women's exposure to environmental tobacco smoke (ETS), and factors associated with husband tobacco use amongst pregnant women, and husband tobacco cessation.  It is important to present theoretical assumptions. Please discuss why those socio-demographic of participants and pregnancy-related factors (such as age, education, employment, gravidity, first pregnancy, childbirth anxiety, and Social support) could affect ETS, husband tobacco use amongst pregnant women, and husband tobacco cessation.  Relevant theories and studies must be used to support the assumptions. 

Author Response: We are grateful to Reviewer 2 for their thought-provoking comment. Accordingly, we have inserted text and cited literature in the introduction section (please see Lines 65-70 and 74-80).

Regarding the methodology, please indicate how those questions used for data collection were developed from. Have you tested reliability and validity of your research tool (measurement)?

Author Response: The questionnaire administered to the participants was curated specifically for the overall Mutaba’ah study using other validated and frequently used questions from similar cohorts around the world including the Avon Longitudinal Study of Parents and Children, Born in Bradford, MoBa, and the Danish National Birth Cohort. The cross-sectional analysis presented in this manuscript was based on the questions in the short questionnaire administered at the start of the study upon enrolment into the cohort. A second longer questionnaire (viz. long questionnaire) is administered between weeks 24-28 when the pregnant women present for gestational diabetes testing (mandatory for all pregnant women in Abu Dhabi) but the data contained in this manuscript is from the baseline measurements upon enrolment into the cohort. Although the long questionnaire includes some psychometrically validated measures with composite scores for various dimensions, the current manuscript uses data from the short questionnaire which uses questions from similar cohorts to collect data on exposures and health status; however, it is not possible to quantitatively test the validity and reliability of these individual questions. Further details on the short questionnaire are available in the published protocol paper cited in this manuscript (ref #12).

Regarding the table 1, please indicate a type of statistic used for testing statistical difference of variable scores. Those statistics are such as t-test, ANOVA, Chi-square, etc.

Author Response: We thank the Reviewer for their comment. Details on the type of statistical test used for different variables is provided in “2.3 Statistical Analyses” section of the Methods “Continuous variables with a normal distribution were compared using the Student’s t-test whilst categorical data were compared using the Pearson Chi-square test. Univariate and multivariate logistic regression models were used to quantify the association between different socio-demographic and pregnancy-related characteristics and maternal/paternal tobacco use and ETS exposure during pregnancy. Backward stepwise multivariate analyses were performed with a removal criterion of p-value equals 0.10.” In addition, we have added brief details of the statistical analyses in the footnote for Table 1.

For the discussion, the interpretations of the results were not clearly. please provide logical reasons why pregnant women who were employed, had childbirth anxiety, and had an increased number of adults living in the same household were i more likely to be exposed to ETS.

Author Response: We are grateful for Reviewer 2’s suggestion. We have reviewed the Discussion section and added further text on the logical reasons why these factors might be associated with ETS exposure in the context of our study with supporting references (please see Lines 272-298).

I also do not understand why pregnant women with higher education levels and higher gravidity were less likely to be exposed to ETS. Please provide concrete reasons. Please also provide reasons how those significant variables could be associated with husband tobacco use amongst pregnant women, and husband tobacco cessation.  

Author Response: Thank you for your comment. An explanation of the relationship between higher education levels, possibly related to health literacy levels, and ETS exposure is provided on Lines 272-278 and we have added text on Lines 275-281 related to plausible explanations for the relationship between gravidity and ETS exposure, and also husband tobacco use (please see Lines 282-298). There were no factors in the multivariate analysis associated with husband smoking cessation.

Reviewer 3 Report

This study is interesting, but requires major revisions:

  1. Please clearly define the study aim in the Abstract section.
  2. The data collection period (2017-2021) may have an impact on the results. The Authors combined data before and during the COVID-19 pandemic, which may have a significant impact on human behaviors toward tobacco use.
  3. Lines 70-71 require language revision 
  4. Please clearly define how the questionnaires were distributed
  5. Table 1 - it is unclear why the Authors divided the population by ETS exposure. How about the smoking status? Please justify this way of data presentation
  6. Please revise the limitations section to address all the potential gaps.
  7. The conclusions section should be more informative
  8. Please provide practical implications of this study (e.g., at the end of the discussion section)

Author Response

Reviewer 3

This study is interesting, but requires major revisions:

Author Response: We thank Reviewer 3 for taking the time to review our manuscript. The insightful feedback has helped us to improve our manuscript and we hope that you think the revised version makes a useful addition to the sparse literature on environmental tobacco smoke exposure amongst pregnant women in the Middle East region.

Please clearly define the study aim in the Abstract section.

Author Response: We thank Reviewer 3 for their comment. We have inserted the study aim in the abstract (please see Lines 21-23).

The data collection period (2017-2021) may have an impact on the results. The Authors combined data before and during the COVID-19 pandemic, which may have a significant impact on human behaviors toward tobacco use.

Author Response: Reviewer 3 has made an insightful comment about the possible influence of the COVID-19 pandemic on temporal changes in parental tobacco use and ETS exposure. Although we are not able to explore this suggestion in our dataset, we have included this point as a future direction for cohort studies with repeated measures of parental tobacco use and/or prenatal tobacco use pre- and during the pandemic (please see Lines 343-348).

Lines 70-71 require language revision 

Author Response: We thank Reviewer 3 for their comment and we have amended the text to improve comprehension.

Please clearly define how the questionnaires were distributed

Author Response: We have added that the questionnaire was completed on an electronic tablet in the Methods section (please see Line 94).

Table 1 - it is unclear why the Authors divided the population by ETS exposure. How about the smoking status? Please justify this way of data presentation

Author Response: The primary research question was related to ETS exposure amongst pregnant women in the UAE. A priori, the analytical plan developed by the research team in the grant (#MBRU-CM-RG2020-08) supporting the study was to create binary exposure groups (Exposed to ETS/Not Exposed to ETS) based on whether the pregnant woman reported ETS exposure at work and/or home.

Please revise the limitations section to address all the potential gaps.

Author Response: We are grateful to Reviewer 3 for their feedback on study limitations. In view of this comment, we have reviewed the limitations section but cannot see which additional limitations have been omitted. We have acknowledged minimising selection bias in the cohort recruitment from the two major public and the largest private hospital in Al Ain, mentioned recall bias, discussed the caveats of our data in relation to self-reported tobacco use compared to objectively assessed tobacco use (e.g. urinary cotinine levels), and highlighted that we used a broad concept of tobacco use. Finally, we also mentioned that the study used baseline cross-sectional data from the cohort study and that the reported associations do not indicate causality.

The conclusions section should be more informative

Author Response: Thank you for your feedback. We have re-written parts of the conclusion to make it more informative (please see Lines 368-374).

Please provide practical implications of this study (e.g., at the end of the discussion section)

Author Response: Thank you for your suggestion. Accordingly, we have added a sub-section on ‘Practical Implications’ (please see Lines 349-361).